# Soluble Compounds Released by Hypoxic Stroma Confer Invasive Properties to Pancreatic Ductal Adenocarcinoma

**DOI:** 10.3390/biomedicines8110444

**Published:** 2020-10-22

**Authors:** Dajia Liu, Anne Steins, Remy Klaassen, Amber P. van der Zalm, Roel J. Bennink, Geertjan van Tienhoven, Marc G. Besselink, Maarten F. Bijlsma, Hanneke W. M. van Laarhoven

**Affiliations:** 1Laboratory for Experimental Oncology and Radiobiology, Center for Experimental and Molecular Medicine, UMC, University of Amsterdam, Cancer Center Amsterdam, 1105 AZ Amsterdam, The Netherlands; dajia.liu@amsterdamumc.nl (D.L.); a.steins@amsterdamumc.nl (A.S.); r.klaassen@amsterdamumc.nl (R.K.); a.p.vanderzalm@amsterdamumc.nl (A.P.v.d.Z.); 2Department of Medical Oncology, Amsterdam UMC, University of Amsterdam, Cancer Center Amsterdam, 1105 AZ Amsterdam, The Netherlands; h.vanlaarhoven@amsterdamumc.nl; 3Oncode Institute, 1105 AZ Amsterdam, The Netherlands; 4Department of Radiology and Nuclear Medicine, Amsterdam UMC, University of Amsterdam, Cancer Center Amsterdam, 1105 AZ Amsterdam, The Netherlands; r.bennink@amsterdamumc.nl; 5Department of Radiation Oncology, Amsterdam UMC, University of Amsterdam, 1105 AZ Amsterdam, The Netherlands; g.vantienhoven@amsterdamumc.nl; 6Department of Surgery, Cancer Center Amsterdam, Amsterdam UMC, University of Amsterdam, 1105 AZ Amsterdam, The Netherlands; m.g.besselink@amsterdamumc.nl

**Keywords:** epithelial-to-mesenchymal transition, hypoxia, pancreatic ductal adenocarcinoma, pancreatic stellate cells, stroma

## Abstract

Pancreatic ductal adenocarcinoma (PDAC) is characterized by abundant stroma and a hypoxic microenvironment. Pancreatic stellate cells (PSC) are activated by hypoxia and promote excessive desmoplasia, further contributing to the development of hypoxia. We aimed to explore how hypoxia and stroma interact to contribute to invasive growth in PDAC. [^18^F]HX4 PET/CT was found to be a feasible non-invasive method to assess tumor hypoxia in 42 patients and correlated with HIF1α immunohistochemistry in matched surgical specimens. [^18^F]HX4 uptake and HIF1α were strong prognostic markers for overall survival. Co-culture and medium transfer experiments demonstrated that hypoxic PSCs and their supernatant induce upregulation of mesenchymal markers in tumor cells, and that hypoxia-induced stromal factors drive invasive growth in hypoxic PDACs. Through stepwise selection, stromal MMP10 was identified as the most likely candidate responsible for this. In conclusion, hypoxia-activated PSCs promote the invasiveness of PDAC through paracrine signaling. The identification of PSC-derived MMP10 may provide a lead to develop novel stroma-targeting therapies.

## 1. Introduction

Pancreatic cancer (PC) ranks fifth in the leading causes of cancer-related death in the Western world [1]. Pancreatic ductal adenocarcinoma (PDAC) accounts for approximately 90% of PC cases [2]. Its incidence is increasing, whereas its outcomes have not improved much [1,3]. Late diagnosis, early metastasis, and limited responses to currently available treatments together result in a dismal prognosis [4]. PDAC is characterized by excessive desmoplastic stroma and hypoxic regions within the tumor mass [2,5]. Both are known to contribute to tumor growth, immune escape, invasion, and resistance to therapies.

Pancreatic stellate cells (PSC) are believed to be essential for normal pancreatic architecture and are the key profibrogenic cells in the stromal reaction surrounding the tumor cells [6,7]. In response to tissue hypoxia, inflammation, or injury, PSCs transform into myofibroblast-like cells [8,9]. The activated PSCs express markers such as α-smooth muscle actin (αSMA), and produce excessive extracellular matrix components (ECM) such as collagen, fibronectin and laminin [7,10,11,12]. In cancer, the balance between ECM production and degradation is perturbed, eventually leading to a dense and fibrotic stroma [10,11,12,13]. The resultant fibrotic tissue compresses vessels, limiting tumor vascularization, thereby further adding to the tissue’s hypoxic status [9,13,14,15,16,17]. The hypoxic milieu leads to hypoxia-inducible factor 1α (HIF1α)-driven processes such as epithelial-to-mesenchymal transition (EMT) [18,19,20,21,22]. EMT promotes tumor cell proliferation, metastatic dissemination, and chemo- and radioresistance, ultimately resulting in tumor progression and relapse in PDAC patients [23,24,25,26,27]. Previous studies have shown the association of hypoxia and stromal activation as explained above, but if and how hypoxia affects tumor biology through the stroma is largely unknown.

The present study aimed to investigate the process through which hypoxia-activated stromal PSCs promote tumor cell invasiveness in PDAC. These findings could aid the development of novel therapeutics that target tumor-promoting stromal programs that rely on hypoxia and HIF1α signaling.

## 2. Experimental Section

### 2.1. Materials and Methods

#### 2.1.1. Patient Cohort

To assess hypoxia in a clinical setting, patients were enrolled from the Amsterdam UMC in two different studies. In the first so-called HYPE study, patients with pathologically confirmed locally advanced or metastatic PDAC were enrolled in the period 05-2012 to 06-2014 (NCT01995084). We previously reported on these patients in our [^18^F]HX4 repeatability study in [28]. If repeated measures were performed, only the first [^18^F]HX4 PET scan was used for the current analysis.

For the second, so-called MIPA study, patients with a CT diagnosis of (borderline) resectable pancreatic cancer (according to the definition of the Dutch Pancreatic Cancer Group [29,30]) were included in the period between 11-2013 and 07-2017 (NCT01989000). Because of a contrast-enhanced MRI scan performed in this study additional inclusion criteria comprised a minimal eGFR of 30 mL/min/1.73 m^2^ and no contraindications to undergo MRI scanning.

Both studies were approved by the institutional review board of the Amsterdam University Medical Centers and written informed consent was given by all patients before the start of the study. Patients underwent [^18^F]HX4 PET/CT scanning before surgical exploration [28]. None of the patients received neoadjuvant therapy. Complete clinical follow-up was used until November 2017. Overall survival was calculated from the date of the [^18^F]HX4 PET scan till dead or last follow-up. Disease-free survival was calculated from the date of [^18^F]HX4 PET till manifestation of local or distant disease or last follow-up. Median follow-up for the patients that were still alive (n = 8) was 41 months (IQR: 31–46). Surgical specimens were used for further immunohistochemical analysis [31].

#### 2.1.2. Image Analysis

Acquired [^18^F]HX4 PET/CT images were viewed side by side with separately acquired diagnostic contrast-enhanced CT images to aid tumor localization. [^18^F]HX4 uptake concentrations values were converted to standardized uptake values (SUV), correcting for injected dose and patient weight. Volumes of interest (VOI) were drawn for both the tumor area and the aorta on the [^18^F]HX4 images. To avoid incorporation of the bile duct, duodenum, or liver accumulated activity in the VOI, the [^18^F]HX4 PET scans were projected on the low-dose CT during VOI delineation. After delineation, tumor to background ratio (TBR) was calculated for all voxels in the tumor VOI by dividing the voxel SUV by the average SUV of the aorta VOI.

From the voxel with the highest TBR within the tumor VOI the maximum TBR value (TBRmax) was defined. In addition, the TBRpeak was defined as the average TBR of the voxel with TBRmax and its 3D 26 connected neighborhood within the tumor VOI. The hypoxic volume (HV) was calculated as the combined volume of all voxels within the tumor VOI with a TBR > 1.4. Tumors with an HV > 1 mL were considered hypoxic. The additional 1 mL threshold was introduced to circumvent image noise and limited resolution of the PET.

As a validation of the [^18^F]HX4 PET/CT results, HIF1α expression was determined on the tissue slice of the 13 corresponding patients with available histopathology (see Immunohistochemistry section below). The stained slices of 11 patients represented a central slice in the tumor and the PET VOI of the entire tumor volume was used for comparison. In two patients, the tissue slices were located at the edge of the tumor, and PET tumor VOI of the entire tumor volume was not representative of the specific location. For these two patients, separately determined tumor VOIs were defined solely on the edge of the tumor, based on the location of the tissue slides from macroscopic photographs taken at pathological examination (see immunohistochemistry). These separate tumor VOIs were only used for the comparison to tissue HIF1α, for all other evaluations the entire tumor VOI was used.

#### 2.1.3. Immunohistochemistry

After fixation, resected tissue specimens were sliced in axial orientation in 5 mm thick slices. All macroscopic slides were photographed to later allow estimating tumor location in the pancreas and to determine the relative location of the selected slice. An entire slice with evident tumor was dehydrated through a series of ethanol, embedded in paraffin and 4 µm thick slices were cut on a microtome. Tissue slices were deparaffinized and rehydrated followed by heat-mediated antigen retrieval using Tris-EDTA buffer solution at pH 9.0 (Lab Vision™ PT module™, Thermo Scientific, Waltham, MA, USA). Hydrogen peroxide (3% in PBS) was used to block endogenous peroxidase and Ultra-V Block (Immunologic, VWR International, Radnor, PA, USA) was used to block unspecific staining. Anti-HIF1-α antibody (610959, clone 54, BD Biosciences, San Jose, CA, USA) was diluted 1:100 in normal antibody diluent (KliniPath, VWR International, Radnor, PA, USA), applied on sections and incubated overnight in a humidified chamber at 4 °C. Slides were incubated for 20 min with BrightVision+ post antibody block, followed by 30 min secondary antibody BrightVision Poly-HRP-anti Ms/Rb IgG (both Immunologic, VWR International, Radnor, PA, USA). After developing the staining with Bright-DAB (Immunologic), sections were counterstained with Haematoxylin Mayer (Klinipath, VWR International, Radnor, PA, USA). Slides were dehydrated and mounted in Pertex mounting medium (HistoLab, Askim, Sweden). Further image processing was performed using custom scripts written in Matlab (R2017b, MathWorks, Natick, MA, USA). On the DAB stained slides, image deconvolution was performed, separating the DAB staining in a separate image channel. Next, this DAB channel was used to automatically determine a threshold in the tumor ROI using the maximum entropy approach. HIF1-α was quantified as the number of DAB-positive stained nuclei per surface area (+/mm^2^) in the tumor ROI.

#### 2.1.4. Cell Culture

Tumor cells including primary cultures AMC-PDAC-053M, AMC-PDAC-067, AMC-PDAC-099 (previously established from PDX tumors [32]), ATCC cell lines PANC-1, Capan-2, MIA PaCa-2, AsPC-1, Hs766T, PSN-1, PANC-89, Capan-1 (ATCC, Manassas, VA) and pancreatic stellate cells (PS1, a kind gift from prof. Hemant M. Kocher, Barts Cancer Institute) were cultured according to standard procedures in Dulbecco’s Modified Eagle Medium (DMEM) supplemented with 8% fetal bovine serum (FBS; Biowest, France), L-glutamine (2 mmol/L), penicillin (100 units/mL) streptomycin (500 µg/mL, Lonza, Basel, Switzerland). Immortalized PS1 cells were kept on selection with puromycin (1 μg/ml, Sigma, Saint Louis, MO). For co-culture, tumor cells and mCherry labeled PS1 cells were plated by the density ratio of 1:2. Once per month mycoplasma tests were performed.

#### 2.1.5. Lentiviral Transduction

Transduction was performed using *pLeGO* constructs with *Venus* (#27340; Addgene, Cambridge, MA, USA) or mCherry (#27339; Addgene, Cambridge, MA). Following 3rd generation lentivirus production in HEK293T, supernatant was collected after 48 h and 72 h, and 0.45 μm filtered (Merck Millipore, Darmstadt, Germany). PS1 cells at approximately 60% confluence were transduced with the harvested virus overnight in the presence of polybrene (5 μg/mL, Sigma, St Louis, MO, USA). Cells were cultured for an additional 3 days before sorting.

#### 2.1.6. In Vitro Treatments

Conditioned medium (CM) of PS1 cells was collected after three days of culturing under normoxia (21% O_2_, 5%CO_2_, and 74% N_2_) for normoxic PS1 supernatant (NPS), or under hypoxia (1% O_2_, 5% CO_2_, and 94% N_2_) for hypoxic PS1 supernatant (HPS), centrifuged at 2000 RPM for 5 min and stored at −20 °C. For indirect co-culture experiments, PDAC cell lines were incubated for 3 days either with NPS (or control DMEM medium) under normoxic culture conditions or with HPS/control under hypoxic culture conditions.

For size exclusion experiments, HPS and unconditioned medium DMEM were filtered over Amicon^®^ Ultra-15 10 kD, 30 kD, 50 kD and 100 kD centrifugal filter devices (Merck Millipore, Burlington, MA, USA) respectively, according to manufacturer’s instructions. PANC-1 cells were exposed to filtered mediums for 4 days under hypoxia. Functional experiments were performed in the hypoxic chamber for 3 days, using NPS/DMEM incubated cells as a negative control and with HPS as a positive control.

#### 2.1.7. Chemicals

For induction experiments, PDAC cells were incubated in NPS with the addition of 10 ng/mL, 50 ng/mL or 100 ng/mL recombinant human IL1α (200-LA-002, R&D systems); for inhibition experiments, PDAC cells were incubated with HPS in the presence of 5μM GM6001 (sc-203979, Santa Cruz, Dallas, TX, USA) and 10% Noggin (produced by stably transfected 293T cells) and 10 µg/mL recombinant human anti-IL1α (AF-200-NA, R&D systems, Minneapolis, MN, USA) respectively.

#### 2.1.8. Transwell Migration Assay

Migration assays were conducted using Corning FluoroBlok cell culture inserts with 8.0 μm pore filters (351152; Corning, Corning, NY, USA). Before the assay, PANC-1 cells were either subjected to the NPS under normoxia or subjected to HPS under hypoxia for 3 days. Cells were washed with PBS, incubated for 45 min in serum-free DMEM with 10 μM cell tracker Green (C2925; Invitrogen, Waltham, MA, USA), washed again, and another 30-min incubation in serum-free DMEM. The lower compartment of the Corning FluoroBlok plate was filled with 600 μL DMEM with/without 1% fetal calf serum (FCS) (as attractant). The upper compartment was seeded with equal amounts of cells per condition. Migration was measured by a cytofluorometer (BioTek Instruments, Winooski, VT, USA) every 2 min for 3 h at the temperature of 37 °C. The migration was controlled for the no-attractant control with serum-free DMEM in the lower compartment.

#### 2.1.9. Flow Cytometry

Flow cytometric staining was performed according to standard protocols [33], using the following antibodies: anti-CXCR4 (1:100, MAB172, R&D systems), anti-vimentin (1:100, sc-73259, Santa Cruz, Dallas, TX, USA), anti-ZEB1 (1:300, HPA027524, Sigma, St Louis, MO, USA), APC conjugated anti-mouse (1:800, 550826, BD Biosciences, San Jose, CA, USA), APC conjugated anti-rabbit (1:800, 4050-11S, Southern Biotech, Birmingham, Al, USA). Propidium iodide (PI, Sigma, St Louis, MO, USA) was used to control for unspecific staining of dead cells. Intracellular epitopes were measured following fixation and permeabilization (BD Biosciences, San Jose, CA, USA; no PI was used). Cells were acquired on a FACSCanto II (BD, Franklin Lakes, NJ, USA). The values represent the geometric mean fluorescence (gMFI) intensity of the appropriate channel corrected for the isotype control, yielding delta gMFI. Data were analyzed with FlowJo v10 (Tree Star, Ashland, OR, USA).

#### 2.1.10. Western Blot

Capan-2 cells were exposed to hypoxia for 0.5 h, 1 h, 2 h, 4 h, with normoxia as a control. After treatment, cells were lysed in RIPA buffer (Cell Signaling, Beverly, MA, USA) with phosphatase and protease inhibitor cocktail added (Cell Signaling). MIA PaCa-2 cells were treated with hypoxia only or hypoxic PS1 supernatant under hypoxia for 48 h, 72 h, 96 h, with normoxia as a control. BCA (Pierce, Thermo Fisher, Waltham, MA, USA) was used to determine protein levels and allow equal loading. Samples were subjected to SDS-PAGE and transferred to PVDF membranes. Membranes were blocked with 5% BSA (Lonza) in Tris-buffered saline with 0.1% Tween-20 (TBS-T). Membranes were incubated overnight at 4 °C with 1:1000 anti-HIF1α antibody (610959, BD Bioscience), anti-ERK1/2 (9102, Cell Signaling), or 1:500 ZEB1 (3396P, Cell Signaling Technology), 1:1000 vimentin (sc-73259, Santa Cruz, Dallas, TX, USA), GAPDH (MAB374/6C5, BioConnect, San Diego, CA, USA); and then probed with horseradish peroxidase (HRP)-conjugated secondary antibodies goat anti-mouse (550826, BD Bioscience) or goat anti-rabbit (4050-11S, Southern Biotech, Birmingham, Al, USA). Both were used at 1:7500. Signals were imaged on a FujiFilm LAS 4000 imager (Fuji, Tokyo, Japan), using Lumi-Light PLUS Western blotting substrate (12015196001, Roche, Basel, Switzerland).

#### 2.1.11. Human Cytokine Array

Human cytokine antibody arrays AAH-CYT-4000 (RayBiotech, Norcross, GA, USA) were performed based on the manufacturer’s protocol using 3-day incubated HPS, with 3-day incubated NPS as a control. The chemiluminescence was captured using a FujiFilm LAS4000 and spot intensity was quantified by ImageJ. The intensity of the chemiluminescent signal was normalized to the positive control that is the controlled amount of biotinylated antibody printed onto the array.

#### 2.1.12. Survival Analysis and Gene Correlations

For gene expression analysis, the R2: Genomics Analysis and Visualization Platform was used (http://r2.amc.nl). Gene expression data files were obtained from 6 independent pancreatic cancer studies. GEO accession numbers are GSE36924 (ICGC, n = 91, data ref: [34], 2012), GSE28735 (Zhang, n = 90, data ref: [35], 2013), GSE62452 (Hussain, n = 130, data ref: [36], 2016), the Cancer Genome Atlas (TCGA, n = 147, data ref: [37], 2017; https://portal.gdc.cancer.gov/projects/TCGA-PAAD), GSE16515 (Wang, n = 51, data ref: [38], 2009), and Bailey et al. 2016 (Bailey, n = 70, data ref: [39]). For TCGA, samples were selected to contain PDAC only. Activated/normal stroma genesets (data ref: Moffitt et al. 2015 [40]) vs. KEGG hypoxia geneset correlation analysis was performed in these 6 datasets above. *HIF1α* and *ACTA2* gene correlation were performed in dataset PDX-PDAC-homologene-Bijlsma-16 (data ref: [41], 2020; RNA-Seq data were deposited at EMBL-EBI ArrayExpress (E-MTAB-6830)). Survival analysis was performed using dataset Bailey.

HIF signaling pathway-related genes were from 4 genesets:(1)“response to hypoxia” (n = 28 genes, from Gene Ontology (GO) project, data ref: [42], 2018, GO:0001666);(2)“reactome regulation of hypoxia inducible factor hif by oxygen” (n = 25 genes, from The Reactome Knowledgebase, https://reactome.org, data ref: [43,44,45,46], 2020, 2018, 2018, 2000);(3)“reactome regulation of gene expression by hypoxia inducible factor” (n = 18 genes) from The Reactome Knowledgebase;(4)“semenza hif1 targets” (n = 36 genes transcriptionally regulated by HIF1A [Gene ID = 3091], data ref: [47], 2001). Overall survival was analyzed by Kaplan–Meier and significance was determined using log-rank statistics. The cut-off was selected by median expression or geneset Z-score.

In survival analysis of the BMP signaling pathways combined with hypoxia signatures, patients were first categorized by median into high/low *BMP* and high/low hypoxia signaling genes expression. Kaplan–Meier was then performed with samples separated by a combination of these two category tracks. Hypoxia signaling pathway used was “semenza hif1 targets”. BMP signaling pathway genes were from the following 5 genesets:(1)“bmp receptor signaling” (n = 226 genes, from Pathway Commons, data ref: [48], 2010; data source: NCI/Nature Pathway Interaction Database, data ref: [49], 2009);(2)“regulation of bmp signaling pathway” (n = 88 genes, GO:0030510);(3)“response to bmp” (n = 167 genes, GO:0071772);(4)“reactome signaling by bmp” (n = 28 genes, data from The Reactome Knowledgebase, data ref: [50,51], 2018, 2010);(5)“signaling by bmp” (n = 19 genes, from Pathway Commons, data source: The Reactome Knowledgebase, data ref: [52], 2004). *p* < 0.05 was statistically significant.

#### 2.1.13. Quantitative Real-Time PCR

In accordance with the manufacturer’s instructions, RNA was isolated (Macherey Nagel, Bioké, Düren, Germany), cDNA was synthesized using Superscript III (Invitrogen, Waltham, MA, USA) and quantitative RT-PCR was performed with SYBR green (Roche, Basel, Switzerland) on a Lightcycler 480II (Roche, Basel, Switzerland). The threshold cycle (Ct) was determined by normalizing values to *B2M*. The primers that were used in this study are listed in Table A1.

#### 2.1.14. Statistics

All statistical analyses were performed using GraphPad Prism 8 or SPSS 10.1 software (SPSS, Inc., Chicago, IL, USA). Error bars show the mean ± S.E.M. For the MIPA cohort, TBRmax, TBRpeak and HV values were tested for normality using a D’Agostino and Pearson omnibus normality test and found not to be normally distributed. These values are reported as median with interquartile range (IQR); TBRmax, TBRpeak and HV were compared between patients with M0 and M1 disease and tested for significance using a two-tailed Mann–Whitney test; Spearman’s rank correlations between TBRmax and TBRpeak were calculated and correlated to the amount of HIF1α positive stained nuclei in the resection specimen. Kaplan–Meier survival curves were drawn for the overall survival (OS) and in the patients with resected tumors for disease-free survival (DFS); maximum discriminating cut-offs were determined for the entire population based on a maximum significance in the log-rank test for overall survival [53]. Best cut-offs were TBRmax 1.64 (1.63–1.65), TBRpeak 1.52 (1.48–1.55), and HV 1 mL; For the in vitro experiments, unpaired Student’s *t*-tests (two-tailed) were used. For cytokine array fold induction, unpaired Student’s *t*-test (one-tailed) was used. A *p*-value of < 0.05 was considered statistically significant.

## 3. Results

### 3.1. [18. F]HX4 Uptake Correlates with Metastatic Disease, Cellular Hypoxia, and Poor Outcome

A total of 42 patients underwent [^18^F]HX4 PET scanning (Figure 1A). In 24 patients, the tumor was not removed due to irresectable locally advanced or metastatic disease determined by preoperative CT or at surgical exploration. In 18 patients, the primary tumor was resected and from 13 of these patients, histology was available. One patient who underwent resection appeared to have small lung nodules on the CT scan before surgery, which turned out to be metastases at follow-up, and was therefore considered M1 stage at the time of the PET scan. Further patient characteristics are shown in Table A2. Comparing [^18^F]HX4 uptake in patients with local and metastasized disease, tumor to background ratios (TBRs) and hypoxic volumes (HV) were found to be higher in patients with M1 disease compared to patients with local (M0) disease, with TBRmax 1.79 (IQR: 1.62–2.05 vs. 1.55 IQR: 1.33–1.63, *p* < 0.001; Figure 1B), TBRpeak 1.66 (IQR: 1.43–1.79 vs. 1.371 IQR: 1.25–1.47, *p* < 0.001) and HV 6.25 (IQR: 1.17–15.93 vs. 0.24 IQR: 0.0–1.19, *p* < 0.001; Figure 1C).

In the 13 patients with available histopathology, the median number of HIF1α positively stained nuclei per mm^2^ was 1011 (IQR: 762–1093). We noted that the signal for HIF1α was strong in the tissue sections with high HX4 uptake, in contrast to the low or negative staining in tissues with low HX4 uptake (Figure 1D). This observation was statistically verified after image quantification and analysis. Spearman’s rank correlation analysis revealed a positive correlation for TBRmax (rho 0.78, *p* < 0.01) and TBRpeak (rho 0.79, *p* < 0.0001) with the number of HIF1α positively stained nuclei on the histological tumor slides (Figure 1E). Given the association between [^18^F]HX4 and cellular hypoxia, we went on to assess whether hypoxia and HIF signaling are related to prognosis.

Survival analysis was conducted, and we found that the median overall survival of the cohort (n = 42) was 14 months (IQR: 7–23). Median DFS in the patients that underwent resection (n = 18) was 15 months (IQR: 10–40). After dichotomizing the entire population based on the best-discriminating cut-offs, we found a significantly shorter overall survival for the high HX4-uptake group compared to the low HX4-uptake group, with a median OS of 9 vs. 21 months (*p* < 0.001) using the best TBRmax cut-off. This was 8 vs. 19 months (*p* < 0.001), by TBRpeak, and 9 vs. 21 months (*p* < 0.001) using HV (Figure 1F). A similar relation between overall survival and HX4-uptake was found in patients that did not undergo a resection: patients with high TBRmax (7 vs. 14 months, *p* = 0.016) and TBRpeak (6 vs. 14 months, *p* = 0.022) and with significant HV (7 vs. 15 months, *p* = 0.012) showed shorter survival. When the population-based cut-offs were applied solely to the patients that underwent resection there was also a significantly shorter OS (Figure 1G) and DFS (Figure 1H) for tumors with high TBRmax or HV. Median OS and DFS were not yet reached for the tumors with low TBRmax or HV. TBRpeak was not able to significantly discriminate between patients with different OS (*p* = 0.011) or DFS (*p* < 0.001).

To validate the association of HIF signaling with patient outcome in other cohorts, we performed survival analysis using the Bailey et al. PDAC gene expression dataset [39]. Samples were dichotomized by the median of HIF1α geneset Z-score, and a significant association with poor outcome was found for high HIF signature expression (Figure A1A–D). These data confirm that HIF signaling at the gene expression level (as a proxy for hypoxia) correlates with poor prognosis, supporting the association with survival of the imaging-based measurements of hypoxia.

### 3.2. Hypoxia-Activated PSCs Promote EMT in PDAC Tumor Cells

To confirm that hypoxia is related with stroma activation, hypoxia pathway genesets and normal/activated stroma genesets were correlated [40]. In six public gene expression datasets, strong correlations were found between the hypoxia and stromal signatures (Table A3, Figure A1E). These associations were particularly strong for activated stroma. We next turned to our previously established dataset of gene expression from patient-derived xenograft (PDX) models for PDAC. The sequential difference between transcripts from the two species allows identification of compartment-specific signaling [54]. Of note, human HIF1α was positively correlated with mouse *ACTA2*/αSMA (R = 0.561, *p* < 0.05). αSMA is a well-established marker for stromal activation, and the correlation indicates a strong correlation between hypoxia (as evident from HIF1α activation in both the tumor and the stroma cells), and stroma activation. Given the strong link between stroma and hypoxia found here, and the previously reported contributions of both to poor outcome, we next proceeded to investigate whether hypoxia and stroma together contribute to invasive tumor growth.

Tumor cells were cultured alone or co-cultured with fluorescently labeled PS1 cells for 96 h under normoxic or hypoxic conditions. Compared to the normoxic PANC-1 co-cultures, the hypoxic co-culture resulted in a shift towards a mesenchymal morphology of the tumor cells (Figure 2A). To characterize this observation in more detail, CXCR4, a marker for invasive and mesenchymal tumor cells was measured in PANC-1, Capan-2, MIA PaCa-2, AMC-PDAC-053M, and AMC-PDAC-067 by flow cytometry [41]. PDAC cells and PS1 populations in co-culture were separated by sorting for mCherry fluorescence (cytoplots shown in Figure 2B). We found that when cells were normoxic, there was no significant difference of CXCR4 expression between monocultured and co-cultured tumor cells; on the contrary, hypoxic monoculture induced expression of CXCR4 in PDAC cells, and a further increase was shown following co-culture (Figure 2C). Of note, this further increase was statistically significant.

To validate the hypoxic status of tumor cells, HIF1α protein levels were assessed by Western blot and the accumulation of HIF1α was observed following hypoxia (Figure 2D). Altogether, these data suggest that hypoxia induces EMT in tumor cells, and that this much increased by signals from hypoxic stromal cells.

### 3.3. Soluble Compounds Released by Hypoxia-Activated PS1 Cells Confer Invasive Growth in Tumor Cells

As we observed that hypoxia-activated PS1 cells promote EMT in PDAC cells, we hypothesized that this could be mediated by soluble molecules released from PS1 cells. To verify this hypothesis, a medium transfer experiment was performed using four conditions: normoxic incubation of tumor cells exposed to control medium (N), normoxic incubation of tumor cells and medium from normoxic PS1 cells (N NPS), hypoxic incubation of tumor cells with control medium (H), and hypoxic incubation of tumor cells with medium from hypoxic PS1 cells (H HPS). PANC-1 and MIA PaCa-2 cells were treated according to these four conditions for 72 h. Strong mesenchymal morphology shifts were observed in H HPS-treated PANC-1 and MIA PaCa-2 comparing to other conditions (Figure 3A). In order to verify this observation, flow cytometry analysis was performed using vimentin (VIM), one of the most frequently used markers for EMT phenotypes [55]. VIM was found to be upregulated in both cell lines when cultured with the H HPS, compared to both N and N NPS groups (Figure 3B). Additionally, this upregulation of VIM in MIA PaCa-2 was also significant compared to H treatment group, confirming that there exist specific contributions to invasive tumor cell phenotypes that derive from hypoxic stellate cells. We also tested whether hypoxic PS1 supernatant could induce EMT in otherwise normoxic PDAC cells (PANC-1 and MIA PaCa-2 cells), and found that this was the case but not statistically significant, which was the case only in hypoxic tumor cells (Figure A2). Considering that NPS in normoxia and HPS in hypoxia are more representative of in vivo settings, we focused on those conditions in which the PDAC cells are also hypoxic. This trend of increased expression of mesenchymal markers (VIM and ZEB1) in MIA PaCa-2 cells was demonstrated by Western blot assay as well (Figure 3C), most notably in the H HPS groups, with a time-dependent tendency. Transwell migration assays revealed an enhanced migratory capacity for H HPS-treated PANC-1 compared to N NPS-treated one (Figure 3D). Together, these data show that soluble molecules are released in the supernatant by PS1 cells under hypoxic stress and that they contribute to tumor cell EMT.

### 3.4. Identification of Hypoxic Stellate-Cell Derived Inducers of Tumor Cell Invasiveness

To determine if the stromal-derived soluble molecules that drive EMT fall in a size range at which most proteins exist, a size-exclusion experiment was conducted. The ability of these supernatants to induce mesenchymal/invasive phenotypes in the tumor cells was determined by exposing tumor cells to filtered supernatants and by subsequently assessing VIM expression by flow cytometry. This demonstrated that the effect was lost from supernatants in which only molecules smaller than 30 kDa were present. In contrast, the effect was largely retained in supernatants containing factors that were >50 kDa and <100 kDa in size (Figure 4A). Having found that the size of the candidate molecule lies between 30 kDa to 100 kDa, we were confident that this molecule is a protein and used a cytokine array to identify it. This unveiled a series of upregulated cytokines released by hypoxic PS1, compared to the protein levels in normoxic incubation (Figure 4B,C). We selected from these the ones with a size that fell within the range determined (indicated in red in Figure 4C) and validated their corresponding mRNA expression by qPCR. IL1α, BMP6, TIE1, MMP10, FGF2 were found to be upregulated under hypoxia at both the protein and mRNA level and were therefore considered candidate proteins responsible for EMT induction (Figure 4D).

To assess the association of the candidate cytokines with patient outcome, we performed a survival analysis on public gene expression data [39]. Samples were dichotomized by median *IL1A, BMP6, TIE1, MMP10, FGF2* expression, and a significant association with shorter survival was only found for the high *IL1A*, and high *MMP10* groups (Figure A3A,B). For *BMP6* we also considered the gene sets that inform on the activation of its downstream pathway, but no significant association with survival was found, also when considered in combination with hypoxia signatures (Table A4). Thus, *IL1A* and *MMP10* were considered further.

To functionally address which of the remaining candidate cytokines is responsible for the PS1-induced EMT, hypoxic PANC-1 and MIA PaCa-2 cells were either incubated with recombinant IL1α in NPS, or with an IL1 inhibitor (anti-IL1α) in H HPS conditions. Although we observed that PANC-1 cells showed a slight induction of a mesenchymal morphology in the presence of recombinant IL1α (not shown), markers remained unchanged (Figure A3C,D). The same outcome was found when we applied recombinant IL1α in additional cell lines including primary lines (Figure A3E,F). The pan-BMP inhibitor Noggin was also used in an H HPS setting, but this did not inhibit EMT induction (not shown), possibly in line with its characteristics causing exclusion mentioned above.

Next, an inhibition experiment was performed using the pan-MMP inhibitor GM6001 to reverse the EMT induction effects seen in H HPS conditions. In the presence of the inhibitor, morphological shifts were indeed prevented in AMC-PDAC-053M cells, as was mesenchymal marker expression in PANC-1, AMC-PDAC-067, and AMC-PDAC-053M (Figure 5A,B; note that each dot indicates a cell line). These results imply that MMP10 is a likely candidate to drive EMT in hypoxic PDAC tumor tissue. This is in line with the known contributions of metalloproteases (MMPs) to invasive growth and the association of such enzymes with mesenchymal tumor biology [56,57]. Given the existence of effective inhibitors of metalloproteases and the specific expression of MMPs in cancer tissue, this could provide a promising lead to develop novel stroma targeting strategies.

## 4. Discussion

The contributions of hypoxia and stroma to PDAC tumor biology have received much research attention in recent years, but their combined contributions to tumor initiation and progression remain underexplored [58]. In this study, we demonstrated that [^18^F]HX4 uptake is a promising method to assess tumor hypoxia in patients and that this hypoxia is associated with metastatic disease and poor prognosis in PDAC patients. Mechanistically, we find that soluble molecules released by hypoxia-activated stromal cells contribute to the invasive growth of pancreatic cancer cells and that among these, MMP10 is likely to be the major driving factor of invasive growth.

We have previously demonstrated the feasibility of visualizing hypoxia in advanced PDAC using [^18^F]HX4 PET/CT [28]. To our knowledge, the study presented here is the first to demonstrate the association between non-invasive measurements of tumor hypoxia in a clinical setting, with the development of metastatic PDAC. Given that this is an (exclusion) criterion for resection, we propose that [^18^F]HX4 imaging could be a valuable tool for treatment decision making. This also holds true for the effective use of novel treatments that act on the stroma and its tumor-promoting signals in particular.

The correlations between [^18^F]HX4 uptake and HIF1α expression at the cellular level confirm previous preclinical work where [^18^F]HX4 tumor uptake was correlated to pimonidazole staining [59]. HIF1α is a direct marker of hypoxia, and its downstream transcriptional programs reflect the cellular response to a hypoxic environment. HIF1α driven pathways have been associated with rapid progression and poor outcome in PDAC [60,61], which is also consistent with our findings: We find that in vitro, EMT induction can be caused by hypoxia alone. However, we also find that this is further increased by the presence of stellate cells, and that a strong correlation between hypoxia and stromal activation exists. Together these findings propose that hypoxia and stroma act in unison to drive invasive tumor cell growth beyond what is achieved by these factors alone.

Our medium transfer experiments show that soluble factors from hypoxic stromal stellate cells drive EMT, and that this does not require cell–cell contact. Among the candidate list we identified (MMP10, IL1α, TIE1, FGF2, BMP6), IL1α has been reported to be released by both stromal cells and PC cells, and to promote tumor growth in PDAC [62,63]. IL1α secreted from tumor cells can promote tumorigenesis in an autocrine manner [64], and stimulate fibroblasts in a paracrine fashion [65,66]. However, we found no evidence that stroma-derived IL1α can lead to EMT in tumor cells or that it is associated with poor prognosis. We proposed that the tumor-promoting effects of IL1α could be a context-dependent phenomenon and may require the presence of an immune system.

Another interesting candidate that we identified but failed to validate as functional, is TIE1 (Tyrosine Kinase with Immunoglobulin Like and EGF Like Domains 1). We observed the upregulation of TIE1, as well as an increase of TIE2 transcription measured by qPCR (data not shown) in hypoxic stellate cells. These proteins are crucial for the remodeling and maturation of tumor vasculature [67]. TIE1 is believed to form a complex with angiopoietins and sustain TIE2 signaling in contacting cells [68,69] and was found increased in several malignancies [70,71,72]. The exact functions of TIE1 in cancer remain unknown, but given the contributions of TIE1 and -2 to angiogenesis, their role in hypoxic tissue seems especially relevant. It is possible that the tumor cells are not the target cell population and that the endothelium should be studied.

This study is the first to identify MMP10 as the main stromal protein to drive EMT in tumor cells. MMPs are proteinases that have often been implicated in cancer metastasis and progression [56]. It has been reported that stellate cells can produce matrix metalloproteinases (MMPs) but also their inhibitors such as TIMP-1, TIMP-2 [8,73]. Our findings show that specifically in the hypoxic setting, MMP10 effects dominate, and that inhibition of MMP10 could be effective against PDACs. Despite disappointing results from clinical studies with MMP inhibitors [74], the development of more effective and specific inhibitors could improve on this. In addition, we propose that patient stratification is required and the presented study suggests that [^18^F]HX4 imaging could provide this.

Several limitations of our study should be acknowledged. First, the prognostic power of [^18^F]HX4 imaging needs validation in an independent cohort with predetermined cut-off parameters. Second, the stroma consists of numerous cell types and even between stellate cells, heterogeneity exists. It is possible that our stellate cell line does not accurately reflect the heterogeneity that exists in this compartment. Likewise, immune cells are absent from our models and in vivo validation would preferable. However, in these models, tight control over oxygen levels is hard to achieve. Third, our cytokine array is by no means genome-spanning and it is possible that possible candidates were not measured. Future work using proteomics approaches could improve on this.

In conclusion, our study showed that tumor hypoxia can be measured non-invasively in PDAC patients and that this strongly associates with metastatic disease and poor prognosis. At the cellular level, we show that this poor outcome can at least in part be explained by the hypoxia-dependent secretion of MMP10 from stromal stellate cells. This then confers tumor cells with a more invasive phenotype. We propose that MMP10 may be a potential future therapeutic target, when used together with [^18^F]HX4 PET/CT for patient selection.

## Figures and Tables

**Figure 1 biomedicines-08-00444-f001:**
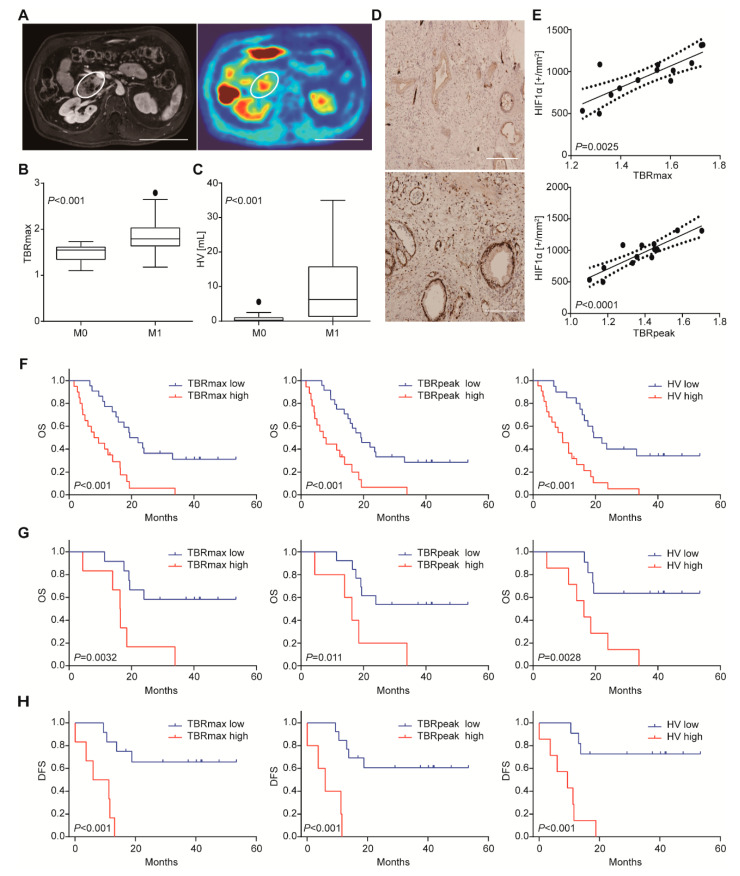
[^18^F]HX4 uptake correlates with metastatic disease, cellular hypoxia, and poor outcome. (**A**) An example of contrast MRI (**left**) of tumor and corresponding PET/CT scan (**right**) demonstrating the high [^18^F]HX4 intake area within the tumor. Scale bar: 10 cm. (**B**) Difference in [^18^F]HX4 TBRmax and HV (panel **C**) between patients with M0 and M1 stage disease. Significance was tested by two-tailed Mann–Whitney test. (**D**) Example of a patient with low (**top**) and high (**bottom**) [^18^F]HX4 uptake and the corresponding HIF1α stained slides of the tumor showing different degrees of cellular hypoxia. Scale bar: 200 µm. (**E**) Correlation between TBRmax (**top**), TBRpeak (**bottom**) and HIF1α positively stained nuclei in the tumor. Kaplan–Meier survival curves discriminating patients based on TBRmax > 1.64, TBRpeak > 1.55 and HV > 1mL. (**F**) Overall survival for the entire population. (**G**) Overall survival for patients where the tumor was surgically resected. (**H**) As for panel G, showing disease-free survival.

**Figure 2 biomedicines-08-00444-f002:**
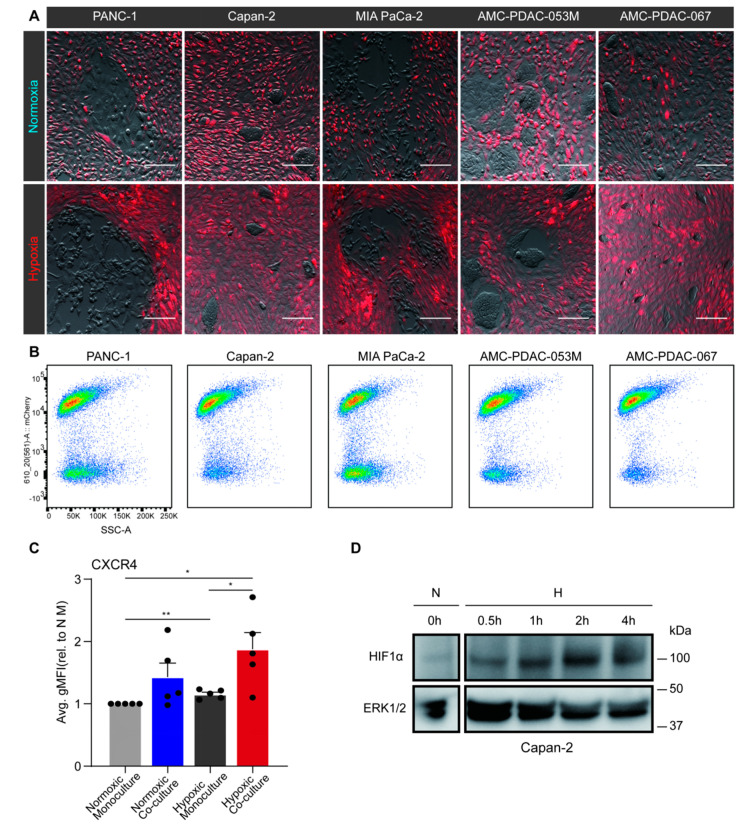
Hypoxia-activated pancreatic stellate cells (PSCs) promote epithelial-to-mesenchymal transition (EMT) in pancreatic ductal adenocarcinoma (PDAC) cells. (**A**) PS1 stellate cells were transduced with an mCherry fluorescent construct (red). Next, untransduced PANC-1, Capan-2, MIA PaCa-2, AMC-PDAC-053M, AMC-PDAC-067 PDAC cells were either co-cultured with mCherry-labeled PS1 or monocultured for 96 h under normoxia or hypoxia. Images are shown of co-cultured cells after 96-h normoxia (upper row) or hypoxia (lower row; brightness and contrast of the images were adjusted). Scale bar: 50 µm. (**B**) Flow cytometry cytoplot examples of the co-cultures as shown in panel A. Populations in the cytoplots indicate mCherry-positive PS1 cells and negative PDAC cells (mCherry signal is on *Y*-axis). Cells were dissociated from the coculture prior to fluorescence-activated cell sorting, separating the PS1 cells and PDAC cells. (**C**) CXCR4 expression of PDAC cells was determined by flow cytometry. Values represent the average gMFI ± S.E.M. of all PDAC cell lines and were normalized to normoxic monoculture group (* *p* < 0.05, ** *p* < 0.01). Each dot refers to a cell line (PANC-1, Capan-2, MIA PaCa-2, AMC-PDAC-053M, AMC-PDAC-067). (**D**) Capan-2 cells were incubated under hypoxic conditions for indicated times with normoxic incubation as a control. Cells were processed for Western blotting, using antibodies against HIF1α. ERK1/2 was used as loading control.

**Figure 3 biomedicines-08-00444-f003:**
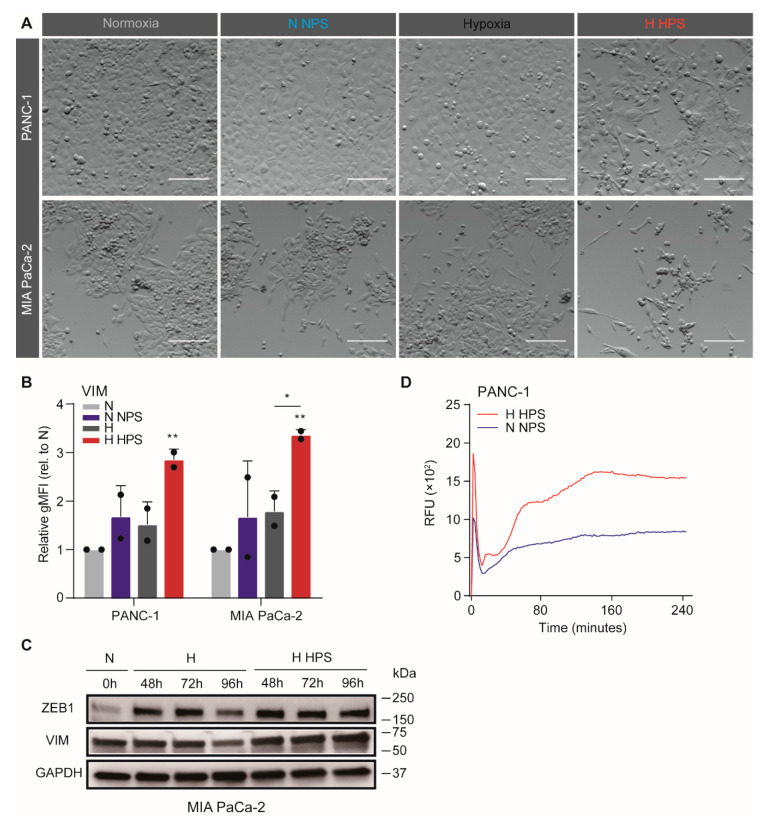
Soluble compounds released by hypoxia-activated stellate cells confer invasive phenotypes in tumor cells. (**A**) PS1 supernatant was harvested after 72-h hypoxia (or normoxia control) to generate hypoxic PS1 supernatant (HPS) and normoxic PS1 supernatant (NPS). Example images of PANC-1 (upper row) and MIA PaCa-2 (lower row) cells exposed to 3 days of NPS/DMEM under normoxia, or with HPS/DMEM under hypoxia (brightness and contrast of the photos were adjusted). Scale bar: 50 µm. (**B**) Expression of VIM in PDAC cells was measured by flow cytometry and values were normalized to normoxic unconditioned medium group. Data show the average gMFI ± S.E.M. (* *p* < 0.05, ** *p* < 0.01). (**C**) MIA PaCa-2 cells were incubated under conditions of H or H HPS for indicated times with normoxic incubation as a control. Cells were processed for Western blotting, using antibodies against ZEB1 and vimentin (VIM). GAPDH was used as loading control. (**D**) Transwell migration assays were performed on PANC-1 cells following treatments with NPS and normoxia, or HPS and hypoxia. 1% FCS was used as a chemoattractant. Migration curves are plotted with S.E.M. and corrected for no attractant controls (medium without FCS).

**Figure 4 biomedicines-08-00444-f004:**
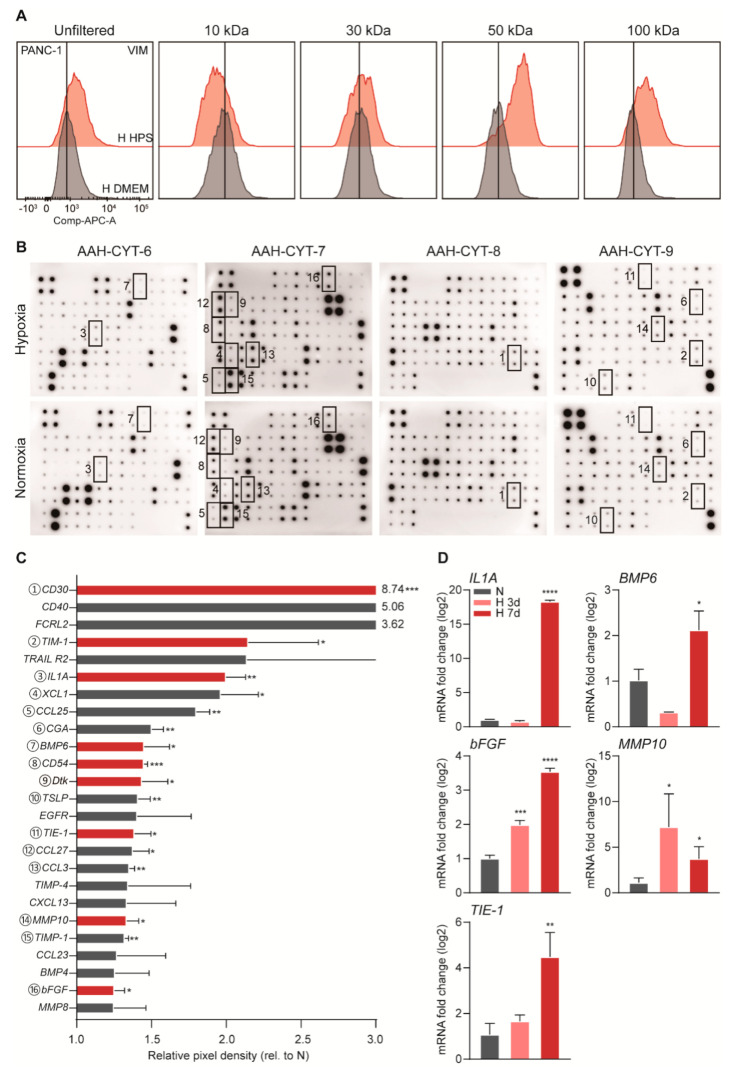
Candidate stromal cytokines upregulated in response to hypoxia. (**A**) PANC-1 cells were exposed to 3 days of unfiltered, 10, 30, 50, 100 kDa filtered HPS (upper row peaks) or unconditioned medium (lower row peaks). Flow cytometry was performed for VIM. (**B**) An AAH-CYT-4000 (RayBiotech, Norcross, GA, USA) cytokine array of four membranes was used to analyze HPS, with NPS as control. Pixel density was determined by measuring spot chemiluminescent signals in ImageJ, and calculated after correction for the negative and positive controls on the membranes and dividing by the NPS control. Coded boxes on membranes scans correspond to matching significantly upregulated cytokines listed in descending order in panel C. (**C**) All highly induced cytokines are plotted. Statistically significant upregulated cytokines within the size range determined (between 30–100 kD) are shown in red bars (* *p* < 0.05, ** *p* < 0.01, *** *p* < 0.001). (**D**) PS1 cells were incubated under hypoxia for 3 days or 7 days, with untreated as control. Candidate cytokine levels were determined by qPCR and normalized to control. Candidates of statistically significant upregulation are plotted (* *p* < 0.05, ** *p* < 0.01, *** *p* < 0.001).

**Figure 5 biomedicines-08-00444-f005:**
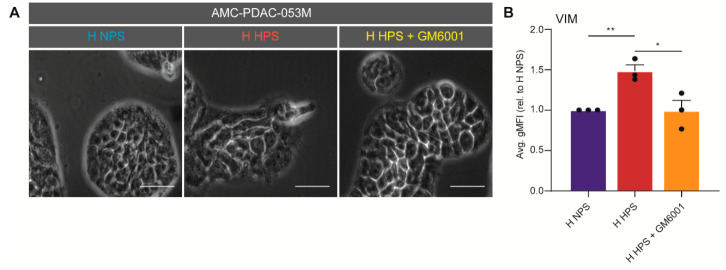
Tumor cell EMT is driven by MMP10. (**A**) AMC-PDAC-053M cells were treated with NPS, HPS, and HPS + GM6001 under hypoxia. Scale bar: 20 µm. (**B**) Expression of VIM in cells were assessed by flow cytometry on PANC-1, AMC-PDAC-053M and AMC-PDAC-067. Data were normalized to NPS group and is shown by average gMFI ± S.E.M. (* *p* < 0.05, ** *p* < 0.01). A dot refers to a cell line.

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
