# Peer review of "Soluble Compounds Released by Hypoxic Stroma Confer Invasive Properties to Pancreatic Ductal Adenocarcinoma"

_biomedicines, 2020, doi:10.3390/biomedicines8110444_

Round 1
Reviewer 1 Report
This is a very elegant and highly relevant study. it will be very interesting to analyse the conclusions in organoid and animal models but this is behind the scope of this paper which should be accepted in the present form
Reviewer 2 Report
In this study Liu D et al. show that hypoxia and HIF1a activation correlates with poor prognosis in pancreatic ductal adenocarcinoma (PDAC). At cellular and molecular level authors claim that pancreatic stellate cells (PSC) release soluble factors in response to hypoxia that subsequently leads to epithelial-to-mesenchymal transition (EMT) in PDAC tumor cells. In this line authors suggest that released MMPs (and MMP10 in particular) by hypoxic PSCs are responsible of EMT phenotype in PDAC tumor cells. Along this line higher MMP10 expression also correlates with poor prognosis in PDAC.
Data are interesting but the following points should be addressed in order to strength the conclusion drawn in this study.
- Authors should silence HIF1 or MMP10 in normoxic and hypoxic PSC cells (PS1 cells) to assess their participation in the EMT phenotype in PDAC tumor cells.
- Authors should characterize better the EMT phenotype in PDAC tumor cells assessing by western blot the expression of several EMT markers such as E-cadherin, vimentin or beta-catenin.
- Authors should assess whether supernatant of hypoxic PSCs induces EMT in normoxic PDAC tumor cells.
- Authors should clarify information in provided in Figure 2B.
Round 2
Reviewer 2 Report
Authors have responded partially my previous comments. Authors propose that pharmacological inhibition provide strong evidence to support the participation of MMP10 in the EMT phenotype induced in PDAC tumor cells. However authors use a pan-MMP inhibitor and therefore this conclusion cannot be drawn without interfering MMP10 specifically in PSC cells. It can be understood those limitations described by the authors to use shRNA constructs but an alternative siRNA approach could be considered. If MMP10 specific silencing is finally not included in the study, authors should only suggest the involvement of MMP10 in the study because its involvement cannot be assured with the current experiments.
However - in line with my previous comments – a better characterization of EMT phenotype in PDAC tumor cells is completely necessary using conventional approaches such as a western blot analysis of several EMT markers such as E-cadherin, vimentin or beta-catenin.
Round 3
Reviewer 2 Report
Authors have responded adequately to my previous comment.